# Neutrophil-to-Lymphocyte Ratio as a Cardiovascular Risk Marker May Be Less Efficient in Women Than in Men

**DOI:** 10.3390/biom11040528

**Published:** 2021-04-02

**Authors:** Ljiljana Trtica Majnarić, Silva Guljaš, Zvonimir Bosnić, Vatroslav Šerić, Thomas Wittlinger

**Affiliations:** 1Department of Internal Medicine, Family Medicine and the History of Medicine, Faculty of Medicine, Josip Juraj Strossmayer University of Osijek, Huttlerova 4, 31000 Osijek, Croatia; ljiljana.majnaric@mefos.hr; 2Department of Public Health, Faculty of Dental Medicine and Health, Crkvena 21, 31000 Osijek, Croatia; 3Faculty of Medicine, Josip Juraj Strossmayer University of Osijek, Huttlerova 4, 31000 Osijek, Croatia; silva.guljas@gmail.com (S.G.); zbosnic191@gmail.com (Z.B.); silvaguljas@gmail.com (V.Š.); 4Department of Clinical Laboratory Diagnostics, Osijek University Hospital, 31000 Osijek, Croatia; 5Department of Cardiology, Asklepios Hospital, 38642 Goslar, Germany

**Keywords:** women, gender, medicine, cardiovascular risk, menopausal transition, meta inflammation, bio-mediators

## Abstract

Cardiovascular disease (CVD) is the leading cause of death in women, although traditionally, it has been considered as a male dominated disease. Chronic inflammation plays a crucial role in the development of insulin resistance, diabetes type 2 and CVD. Since studies on women were scarce, in order to improve diagnosis and treatment of CVD, there is a need to improve understanding of the role of inflammation in the development of CVD in women. The neutrophil-to-lymphocyte ratio (NLR) is an inexpensive and widely available marker of inflammation, and has been studied in cardio-metabolic disorders. There is a paucity of data on sex specific differences in the lifetime course of NLR. Men and women differ to each other in sex hormones and characteristics of immune reaction and the expression of CVD. These factors can determine NLR values and their variations along the life course. In particular, menopause in women is a period associated with profound physiological and hormonal changes, and is coincidental with aging. An emergence of CV risk factors with aging, and age-related changes in the immune system, are factors that are associated with an increase in prevalence of CVD in both sexes. The aim of this review is to comprehend the available evidence on this issue, and to discuss sex specific differences in the lifetime course of NLR in the light of immune and inflammation mechanisms.

## 1. Differences between Men and Women in Cardiovascular Disease Expression and Cardio-Metabolic Pathways

### 1.1. Differences between Men and Women in Cardiovascular Disease Expression

Despite significant progress in its prevention and treatment, cardiovascular disease (CVD) remains an important cause of morbidity and mortality worldwide, in particular in developed countries [1]. Although atherosclerotic CVDs, such as coronary artery disease (CAD) and ischemic stroke, can develop in middle age, CVD appears at higher rates in older age groups (>60) and are associated with high mortality rates or disability [1]. Giving that population aging is a global trend, CVD has a large burden on healthcare systems and societies [2]. There is a need to develop strategies for identifying high risk patient groups more precisely, and for providing more personalized prevention.

Our understanding of the pathophysiology of CVD, and of its environmental and behavioral risk factors, has greatly increased in the last decades [3]. Yet, some aspects of CVD are still poorly understood. This includes, i.e., the mechanisms underlying differences between men and women in the expression of CVD [4]. Although evidence indicates differences in CVD patterns between men and women, it is difficult to distinguish the extent to which these differences are caused by biological (sex-related) factors, such as sex hormones and genes encoded by sex chromosomes, or by social and cultural (gender-related) factors, which could influence differences in lifestyle, the levels of medical care received, and compliance with treatment [5,6].

Current epidemiologic data indicates that CVD is the main cause of death in women, as has already been reported in men [1]. However, the life-course dynamics of CVD differ significantly between men and women. Before menopause, women are relatively protected against CVD, while after menopause, their risk for CVD increases sharply, equaling that of men at the age of 70 [7]. In addition, there are differences in the presentation of CVD. In women, ischemic heart disease (IHD) is less likely to be caused by anatomical obstruction of the coronary arteries, as is the case in men, but instead by disorders such as abnormal coronary reactivity, microvascular endothelial dysfunction, and micro-embolism due to the erosion of active atherosclerotic plaque [7]. Accordingly, IHD in women presents with atypical symptoms and signs. This makes the diagnosis of IHD challenging and may be a reason for women receiving less intensive treatment and having worse outcomes for CVD than men [8]. Furthermore, heart failure with preserved ejection fraction (HFpEF) affects women disproportionately. A typical clinical profile of a woman with HFpEF includes an older age, hypertension, and obesity or diabetes mellitus type 2 (DM2). The mechanisms underlying this disorder are poorly understood. The prevalent theory is that its hallmarks, cardiac remodeling and fibrosis, are the result of microvascular endothelial dysfunction-related chronic tissue inflammation [9].

### 1.2. Differences between Men and Women in Cardio-Metabolic Pathways

The differences between men and women in patterns of CVD can be connected with differences in the life-courses of major CV risk factors, including hypertension, metabolic syndrome (MS) and DM2. MS is a cluster of disorders that build on the foundation of abdominal obesity, such as insulin resistance, hypertension, impaired glucose tolerance, and abnormal HDL-cholesterol and triglyceride levels [10]. This syndrome is known to increase the risk of both DM2 and CVD.

It is well known that hypertension and obesity are closely related to each other, with abdominal obesity being more important than general obesity [11]. In contrast to premenopausal women, who are characterized by general obesity (due to the accumulation of subcutaneous fat), men and postmenopausal women are characterized by abdominal obesity. Accordingly, hypertension already starts to emerge in men after adolescence and in middle age, with rates increasing steadily with age. In women, the critical period in which hypertension starts to emerge, is menopause. Because weight gain in postmenopausal women is of the abdominal type, hypertension appears as part of MS rather than as an isolated disorder [12].

The life-course differences between men and women in patterns of DM2, are similar to those for MS. While men have a significantly higher prevalence of DM2 in middle age, the rates of DM2 increase for both sexes after the age of 55 and become fairly similar in later life (about the age of 70) [13].

## 2. Proposed Underlying Mechanisms of Differences between Men and Women in Cardiovascular Disease Patterns

### 2.1. The Effects of Variations of the Female Sex Hormones

The differences in CVD patterns that have been observed between premenopausal women and age-matched men, and between premenopausal and postmenopausal women, have traditionally been explained by changes in the levels of the female sex hormones, estrogens. Estrogens are known to have cardio-protective effects in premenopausal women, while their decline, during menopause, is associated with marked increase in prevalence of CVD in postmenopausal women.

These changes in the female CV risk profile, after the age of 50 (the average age of menopause), are difficult to distinguish from age-related changes in CV risk factors [14]. Changes in body shape and composition, which occur with aging, including loss of muscle mass and increase in fat tissue, in particular visceral fat tissue, may exaggerate inflammatory response and induce peripheral tissue resistance to insulin, predisposing older individuals to metabolic disorders, hypertension, and the development of the structural changes of the CV system [15,16].

Some other physiological changes that affect women during menopause, such as emotional fluctuation and sleep disturbances, which are thought to result from both hormonal and psycho-social factors, can also contribute to worsening of the cardio-metabolic profile in postmenopausal women. These factors are thought to operate through mechanisms such as behavioral changes, weight gain, activation of the neuro-endocrine stress axis, and increased inflammation [17].

### 2.2. The Effects of Sex Specific Differences in Immune Responses

The direct protective effects of estrogens on the vasculature have been demonstrated in many experimental studies [18]. There is an increasing interest of the researches on the effect of estrogens, and of the sex hormones, in general, on immune cells and immune reactions. This increased interest in immune mechanisms of CVD is associated with the current understanding of atherosclerosis and CVD as a part of inflammaging [19]. This is a concept which explains the development of the common aging diseases, including also DM2, Alzheimer’s dementia, osteoporosis, and some types of cancer, as an integrative part of the aging process, with inflammation as the main driving mechanism.

In this regard, the evidence suggests that estrogens are likely to promote polarization of the immune reaction towards a humoral (antibody-mediated) immune response, in which the main role has the Th2 (helper) and Treg (regulatory) types of CD4+ T lymphocytes. In addition, the effects of estrogens have been demonstrated on increased production of antibodies by B lymphocytes [20]. These characteristics of the immune reaction in women is a mechanism for avoiding a more dangerous cytotoxic (Th1) cell-mediated reaction, which could be directly opposed to women’s reproductive function (Figure 1) [21]. These immune mechanisms may help explain why premenopausal women are, on the one hand, protected against atherosclerosis (a predominantly cell-mediated pathological process), and on the other hand, predisposed to autoimmune diseases (ADs) (a pathological process driven by auto-reactive antibodies) [20,22].

In contrast to estrogens, androgens direct the immune reaction towards the predominantly cell-mediated (Th1) responses, contributing to the male predisposition to the development of atherosclerotic vascular lesions (Figure 1) [22]. This modulation of the immune system by androgens, when combined with early exposure to the behavioral factors such as smoking, or the early onset of hypertension and MS, may be behind the earlier onset of CAD in men than in women, even in middle ages (40–60 years) [21,22].

Recent findings implicate also an important role of the genetic factors in these observed sex specific differences in immune reaction. In this respect, it has been recognized that the X-chromosome encodes about 10 times as many genes as the Y-chromosome and that most of these genes are involved in immune responses [20]. Through the mechanism of the X-chromosome transcriptional silencing in somatic cells, some variants of immune-related genes can escape inactivation, acting towards the disruption of the regulation of self-tolerance and in favor of the development of ADs [23]. This process is associated with turning the balance from the dominance of Th2/Treg (tolerogenic) to the dominance of Th1/Th17 (auto-immune/inflammatory) responses [24]. The final result is the increased production of pro-inflammatory cytokines of the IL-17A type, that are known to be responsible for mediating chronic tissue inflammation (Figure 1).

The dominantly expanded Treg cells, as opposed to the effector Th1 and Th17 cells, is a safeguard mechanism for maintaining immune tolerance and against establishing the devastating effects of autoimmune reactions [23]. Treg cells was found to expand during pregnancy, ensuring tolerance for the semi-allogenic fetus, which justifies this mechanism in females as being evolutionary important [25]. On the other hand, the Treg cell compartment must be very flexible and allow for the alternate dominance of the regulatory and the effector immune functions, to ensure the immune system to provide protection against infections (Figure 1) [26].

Situations in which the risk of the disruption of this mechanism is increased, include periods in women’s life when significant variations in the levels of sex hormones take place, such as puberty, pregnancy, and menopause [27]. The consequence is unresolved chronic inflammation which can lead to the development of ADs and the progression of other chronic inflammatory conditions. In particular, in post-menopause, when age-related tissue damage, due to the emergence of cardio-metabolic disorders, such as hypertension, MS, and DM2, deliberates a large amount of self-antigens, the breakdown of immunologic tolerance can lead to accelerated atherosclerosis and target organ damage (Figure 1) [22,28].

Although ADs typically appear at younger ages, another peak is during early post-menopause (late onset ADs) [27]. In this case, ADs appear at lower rates and with less intensive disease activity. However, because of the pre-existing microvascular endothelial dysfunction and end-organ damage, due to the emergence of cardio-metabolic disorders, these late onset ADs are associated with greater complications [27,28].

### 2.3. The Effects of Age-Related Changes in Immune Mechanisms

Aging is associated with an increase in the level of systemic inflammation, which is considered as an engine in the development of age-related chronic diseases, and CVD, in particular [29]. The sources of pro-inflammatory cytokines, including tumor necrosis factor alpha (TNF-α), interleukin-1 beta (IL-1β), IL-6, IL-12, IL-18, and interferons type 1 (IFNs I), are numerous, and are associated with cell senescence and aging of the immune system.

In general, there is a bias towards non-specific and innate immune responses [30]. Chronic activation of innate immune cells (dendritic cells and macrophages), due to a lifetime long exposure to exogenous antigens, and to increased stimulation by self-antigens, which occurs as a result of increased tissue damage, is considered as the mechanism which contributes the most to increased systemic inflammation.

In particular, the mechanisms such as turning the bias towards autoimmune reactions (with a simultaneous decline in mounting immune reactions to exogenous antigens), defective clearance of apoptotic cells by macrophages, and a continuous activation of the Th1/Th17 pro-inflammatory immune pathway, are considered responsible for sustaining chronic inflammation and continuous tissue damage (Figure 1) [26,30,31,32]. These reactions are associated with increased neutrophil recruitment to the tissue and their prolonged persistence in inflamed tissue [33].

In this context, neutrophils are also considered as the major effector of tissue damage in both atherosclerosis and CV risk factor-associated end-organ damage, where dysfunctional endothelial cells allow for increased neutrophils’ intravascular activation and their translocation to tissues [34,35]. The role of neutrophils in tissue damage is associated with many tissue devastating mechanisms that are inherent to their function, including release of a large amount of pro-inflammatory cytokines, proteolytic and oxidizing enzymes, and reactive oxidative species (ROS) [36]. A specific pro-inflammatory mechanism of neutrophils is also a creation of neutrophil extracellular traps (NETs)—decondensed chromatin products, which can greatly augment inflammation (Figure 1).

In inflamed tissue, there is an increased production of the cytokine IL-23 from innate immune cells. This cytokine is the upstream driving cytokine in the IL-23/IL-17 axis, which is a driving mechanism in the expansion of Th17 lymphocytes (Figure 1) [24]. In addition to Th17 lymphocytes, in aging immune system, the expanded non-specific and innate immune cells also become the source of the cytokine IL-17A, thus amplifying the effects of this cytokine on maintaining chronic inflammation and mobilization of neutrophils to the inflamed tissues [33].

The unrestricted expansion of Treg cells, as a result of insufficient compensation for effector T cell proliferation, can lead to increased production of anti-inflammatory cytokines, such as transforming growth factor beta (TGF-β), which is included in processes of tissue remodeling and fibrosis (Figure 1) [37].

In postmenopausal women, these age-related changes in the immune system coincidence with the decline in estrogen levels and the loss of estrogens’ effects on sustaining immune regulatory functions [21]. These regulatory functions include keeping up an anti-inflammatory phenotype of macrophages and the dominant activity of the Th2/Treg (tolerogenic) immune pathway, over the effector (inflammatory) pathways. By taking a view from the evolutionary perspective, it can be said that gene variants that had a beneficial role in women during their reproductive period, after menopause lose this role, which is associated with turning the bias from the dominance of Th2/Treg (tolerogenic) to Th1/Th17 (autoimmune/pro-inflammatory) immune responses [38].

The pre-existing tissue damage, due to end-organ disease in cardio-metabolic conditions, which emergence is taking place at age 55–60, can further amplify all of these mentioned mechanisms [30]. In this regard, the evidence indicates that both hypertension and MS, as well as DM2, are associated with microvascular endothelial dysfunction [39]. The increased adhesion of activated leukocytes at the surface of the dysfunctional endothelium in the microcirculation, and their increased recruitment from the circulation to tissue, is considered a paramount of the development of end-organ damage. It is also considered as a driving mechanism in the development of atherosclerosis in large arteries [40]. In relation to hypertension, the activated IL-23/IL-17A immune axis has been found to cooperate with angiotensin II (Ang II)—the principal effector of the tissue renin-angiotensin system—in augmenting tissue inflammation and end-organ damage through mechanisms such as mobilization of immune cells to tissue, production of ROS, and tissue fibrosis (Figure 1) [37].

## 3. Leukocyte Counts and Ratios as Markers of Inflammation in Cardiovascular Disease

Low grade inflammation has been implicated in the development of DM2, CVD and other common aging diseases [29]. This has led to various inflammatory markers being studied extensively, for prognostic purposes, in the context of aging diseases and mortality. The most commonly used inflammatory markers were IL-6 and C-reactive protein (CRP) [41]. This early trend has now changed to the more prevalent assessment of the total white blood cell count (WBC) and specific leukocyte classes, as being less expensive and more easily accessible markers of inflammation. The total WBC count comprises several cell types, including granulocytes (mainly neutrophils), monocytes and lymphocytes. When monocytes enter the circulation and come to tissue, they serve as precursors of macrophages or dendritic cells (DCs) [42].

Inflammation is associated with increased recruitment of inflammatory and immune cells from the circulation to the tissue via dysfunctional vascular endothelial cells [35] (Figure 1). Monocytes/macrophages and T-lymphocytes are cell types that are prevalent in atherosclerotic plaque. Growing evidence indicates also the role of neutrophils in both, the development and the progression of atherosclerotic lesions, and plaque destabilization and rupture. Emerging evidence indicates the critical role of neutrophils in CV risk factor-related target-organ disease (Figure 1) [34]. This role of neutrophils is associated with their increased recruitment from the circulation by dysfunctional endothelium and prolonged persistence in tissue, due to decreased apoptosis and disturbed clearance by macrophages [35,43].

The role of lymphocytes is to mount specific (adaptive) immunity [44]. The evolution of the specific immune reaction begins with antigen presentation by DCs to naive T helper (CD4+) lymphocytes (Figure 1). Through the process of proliferation and differentiation of specific T cell clones, there is a parallel process of effector functions polarization, towards either a predominantly humoral, T helper 2 (Th2)-mediated, or cellular, Th1-mediated immune response, depending on which cytokines are dominant in the micro-environment (Figure 1). A key mechanism that protects tissue against prolonged immune reactions or immune reactions to self-antigens, is the active suppression of effector T cell functions by regulatory T cells (Treg) (Figure 1) [45]. In conditions associated with tissue damage or hypoxia, there can be a non-resolving inflammation and/or auto-immune reactions, by redirecting the immune reaction towards the dominance of the Th1/Th17 effector pathway (Figure 1) [46].

The total WBC has been recognized as being associated with CVD and predictive of specific CV and overall mortality [47]. However, many factors have been found to contribute to variations in this parameter, for example, age, sex, smoking, clinical markers of obesity and insulin resistance, which places limitations on its use as a predictive marker in CVD [47]. Similar limitations on the prognostic value of major types of circulating leukocytes, have also been described. In order to account for interactions between different leukocyte types in predicting mortality and other outcomes, which suggests that particular leukocyte types play different roles in the pathophysiology process, leukocyte subtype-based ratios, such as neutrophil-to-lymphocyte ratio (NLR), lymphocyte-to-monocyte ratio, or platelet-to-lymphocyte ratio, have recently begun to be used as prognostic factors in patients with cancer or CVD, or to predict specific and all-cause mortality in the general population [48].

The left side—cell-mediated (Th1 type) immunity is the key immune mechanism in atherosclerosis. Participating cells include macrophages, NK (natural killer) cells, dendritic cells, CD4+ Th_1_ (T-helper, type 1), T-lymphocytes, and CD8+ (cytotoxic) T-lymphocytes. The immune reaction is initiated by the interaction of toll-like receptors (TLR), exposed on the surface of dendritic cells, with antigens. Activated dendritic cells release the cytokine IL-12, which together with the cytokine interferon-gamma (IFN-γ), a product of activated CD4+ (Th1) lymphocytes and NK cells, has a key role in the activation and recruitment of cells, protagonists of cell-mediated immunity. Activated macrophages with a pro-inflammatory phenotype, phagocyte oxidized LDL-cholesterol form the foam cells. Activated macrophages contribute to tissue damage, necrotic lipid core formation, chronic inflammation, and plaque instability by producing different soluble mediators, such as metalloproteinases.

The right side–immune mechanisms responsible for the development of multiple-organ disease, as found in cardio-metabolic conditions, including hypertension, metabolic syndrome and diabetes type 2, and in systemic autoimmune disease. Immune cells transmigrate through the dysfunctional endothelial cells of the microcirculation. The specific immune reaction is initiated by the interaction of activated dendritic cells with naïve, helper (CD4+) type T lymphocytes. The immune reaction develops either towards a humoral (Th2) or cell-mediated (Th1) immune response. Tissue damage or hypoxia create an environment for unresolved chronic inflammation, tissue remodeling and fibrosis. It arises from the imbalance between regulatory immune functions, presented with Treg (regulatory T lymphocytes) and anti-inflammatory cytokines, IL-10 and transforming growth factor beta (TGF-β), and effector immune functions, presented with Th17 lymphocytes, and pro-inflammatory cytokines, IL-23, IL-6, and IL-17A. The established IL-23/IL-17A immune pathway, via the production of Granulocyte and granulocyte macrophage colony stimulating growth factors (G-CSF and GM-CSF), is responsible for the continuous recruitment of activated neutrophils to tissue (increased granulopoiesis). Neutrophils, working in concert with the activated renin-angiotensin system and angiotensin II (Ang II), are the main protagonists of tissue damage and fibrosis.

## 4. Neutrophil-to-Lymphocyte Ratio (NLR) as a New Cardiovascular Risk Marker

The NLR is calculated from the complete blood count with differential, and is an inexpensive, easy to obtain, and widely available marker of inflammation, with a predictive value comparable to that of CRP. This marker can be used to improve risk stratification in patients with various CVD and cardio-metabolic conditions [49]. A number of researchers concluded that NLR is a more powerful predictor than the total WBC or any of leukocyte subtypes. A reason for this could be that NLR is a ratio of two opposite but complementary immune pathways. On the one hand, it reflects the effect of the neutrophils, that are responsible for nonspecific immune response in inflammation. On the other hand, it reflects the role of lymphocytes, as the critical players in specific immune response. Furthermore, compared to WBC, NLR is much less influenced by physiological conditions, such as physical training or bodily dehydration [49].

In comparison to the effect of each single component of NLR on survival, the reports are consistent in findings that an increased neutrophil count is associated with lower survival, while the situation is more complex in the case of lymphocytes. Namely, reports indicate associations of both, higher and lower lymphocyte counts (lymphocytopenia), with worse outcomes, which depends on the clinical contexts, or there are no reports on these associations at all [50].

Since CVD is a leading cause of death, there is a great interest in strategies for detecting high risk patient groups, by screening in the population [51]. The main cause of these diseases is atherosclerosis. Carotid intima-media thickness is generally accepted as an atherosclerosis stratification risk marker; it correlates well with coronary atherosclerosis and can predict CV events but requires imaging diagnostic methods [52]. A number of studies have identified NLR as an inflammatory marker with good prognostic value in CVD (Table 1). Elevated values for this marker were shown to be associated with CAD and acute coronary syndrome (ACS) and their outcomes and have been reported to predict outcomes in patients undergoing coronary artery revascularization interventions [51]. The NLR has also been shown to be a reliable predictor of short- and long-term mortality in acute cerebrovascular incidents.

In the number of studies, it has been indicated that NLR has a great potential as an easily available laboratory marker for a large number of cardio-metabolic conditions that carry an increased risk of CV and cerebrovascular events (Table 2). However, many of them suggest that their results should be confirmed by further research, to confirm the value of NLR and to better define its role in everyday clinical decisions.

Use of the NLR has also been proposed to improve the diagnosis of chronic conditions other than CVD, the monitoring of disease activity, and the prediction of outcomes, in areas such as malignant diseases, ADs, mental and neurological disorders, such as major depressive disorder, schizophrenia and Parkinson’s disease, supporting the theory of inflammation activation in these conditions, with NLR potentially serving as a marker of inflammatory activity [60].

Despite the growing evidence indicating NLR as a marker that could make a significant change to everyday clinical practice, its implementation in routine practice is still a challenge. This is due to the need to adjust NLR values to patient demographics and health-related factors. Namely, although high NLR values are associated with poorer clinical outcomes, the full range of factors that influence the magnitude of the NLR value are poorly understood. Using data from the NHANES (National Health and Nutrition Examination Survey) survey, Howard et coll. have recently shown that multiple demographic and lifestyle factors are associated with NLR, and independently of important comorbidities, including heart disease, cancer, DM2, and hypertension [67].

Problems that also need to be solved before NLR can be used routinely, are in determining a reference range in a population of healthy people, depending on age and sex, and in determining the thresholds for predicting poor outcomes or adverse disease courses in particular diseases. The study by Forget et al. evaluated NLR in an adult population (22–66 years old) in Belgium, free of acute or chronic diseases, with the aim to determine the reference values, and suggested a normal reference range of 0.78–3.53. Researchers from the Rotterdam study reported a mean value of NLR and the corresponding 95% reference intervals of 1.76 (0.83–3.92), for the general population old 45 years and more [68]. It was shown in this study that NLR increases with age and is higher in males than females.

## 5. Sex Specific Differences in NLR across the Life Course–Associations with Immune Mechanisms Contributing to Sex Specific Differences in Patterns of CVD

Although it is well accepted that there is a sexual dimorphism in patterns of CVD, in particular with respect to differences in age (<60 vs. >60 years), and that NLR has a potential to be a powerful predictive marker in CVD, no studies, designed specifically to demonstrate sex bias in the utility of NLR in CV risk prediction, have been performed to date. One of the reasons for this gap in research could be a spurious inference that might have arisen from population-based studies with retrospective design, showing that an increase in NLR value generally indicates worse survival, independent of age, sex, or comorbidity status [69]. In addition, except for the evaluation of how each of the two components of NLR, neutrophil and lymphocyte counts, influence the magnitude of NLR, there have been no studies that have taken immune mechanisms, that are likely to impact these parameters’ values, into account. Thus, it is not possible to determine from the NLR value, whether it reflects an inflammatory state, as indicated by increased neutrophil count, or a depressed host immune competence, as indicated by increased lymphocyte count.

It has increasingly been realized that there are differences in the peripheral blood leukocyte composition and count ratios, both between men and women and between women before and after menopause. In premenopausal women, differences in reproductive history such as the age at menarche and menopause, and the number and time of births, were found to influence NLR values [70]. Studies including a lifetime perspective of these parameters are scarce.

Based on two such published studies, carried out on a large cohort of healthy Chinese individuals (free from CVD), it was possible to reconstruct the sex specific life course of NLR and its components—neutrophil and lymphocyte counts (Figure 2) [71].

Overall, a comparison for women showed that premenopausal women (age < 50) have higher neutrophil counts and percentages, and NLR, than postmenopausal ones (age 51–70), with the drop of NLR occurring at the time of menopause (around age 50) and in early post-menopause (age 50–60).

With respect to sex specific differences, it was shown that in the period of life before menopause (<50), women have higher neutrophil counts and percentages, lower lymphocyte counts and percentages, and higher NLR, than age-matched men. With increasing age (50–60), NLR was shown to increase steadily in men, due to increased neutrophil counts and percentages, and decreased lymphocyte counts and percentages, while in women, there is a menopausal drop in NLR. Differences in NLR between men and women begin to vanish at ages above 60, due to a faster rise in neutrophil counts and percentages, and slower decline in lymphocyte counts and percentages in women, in comparison to men.

Similar lifetime courses were also found in the South Korean population [72]. In this study, for ages below 50, NLR was shown to be higher in women than in men, while this was reversed for ages above 55. In addition, there was a menopausal fall in NLR, being influenced by a decrease in neutrophil count and percentage, while the lymphocyte count and percentage did not change significantly.

It is important to keep in mind when reconstructing the sex specific lifetime course of NLR, that there are racial differences in the average and cut-off values of NLR, with lower values being indicated for the Asian compared to the European populations, although the patterns of change are likely to be the same [73].

As they were indicated in two studies mentioned above, we used sex specific differences in the NLR lifetime course to discuss possible immune mechanisms that could have an influence on these differences, and sex-related differences in CVD patterns. A better understanding of these associations could speed up research and improve utilization of NLR in prediction of CVD-related outcomes.

Below the age 50 (pre-menopause), the mean NLR values are higher in women than in men and show an oscillating course. While in men, the mean NLR values steadily increase after the age 50, in women, there is a sudden drop in NLR values around the time of menopause (50–55 years), followed by the recovery of the NLR slope in later life (after the age 60). Frail older women have a steeper NLR slope and catch up with men at the age 70.

### 5.1. Sex Specific Differences in NLR in the Age of Pre-Menopause

In the period of life before menopause, women have a stronger inflammatory response than age-matched men and are better protected against infections [74]. This is evident from a faster increase in blood neutrophils during acute inflammation, but it is counteracted by a faster inflammation resolution [43]. The net effect is that women have more preserved endothelial function in response to inflammation, which helps explain why premenopausal women are better protected against CVD than middle-aged men [18].

This characteristic of female immune reaction arises from the effect of estrogens on immune cells and immune functions. These effects include increased recruitment of neutrophils from bone marrow to blood, and their increased resistance to apoptosis, which makes that women respond faster and stronger, than men, to inflammatory challenges [21]. In addition, estrogens act in a way to support activation of B lymphocytes and their transformation into antibody-producing plasma cells, which coincidences with the characteristic of women to mount higher levels of antibodies on inflammatory stimulus. This type of immune reaction, which includes a prominent role of neutrophils and antibody-mediated immune responses, is also supported by the female genetic predisposition for higher propensity of the IL-23/IL-17A immune pathway, known as being in the background of their predisposition to ADs [20,24].

That women in their reproductive phase of life respond with higher level of inflammation but are relatively protected against the development of atherosclerosis, this can be explained by this specific genetic background, and by the proposed effect of estrogens on polarization of immune reaction towards the tolerogenic (Treg) and humoral (Th2) immune responses [21]. An anti-inflammatory effect of estrogens on macrophages can also contribute to the protection against atherosclerosis, giving the fact that macrophages are the main protagonist of cell-mediated (Th1) immune reaction and vascular inflammation associated with the development of atherosclerotic lesions [75].

On the contrary, the action of testosterone was found to be in potentiating the development of atherosclerosis, by mechanisms such as polarization of immune reaction towards cell-mediated (Th1) responses, and inhibition of antibody production by B lymphocytes [21]. This characteristic of immune reaction makes that middle-aged men are more prone, than premenopausal women, on the development of atherosclerotic CVD.

Because of these sex specific differences in immune reaction, higher NLR, in premenopausal women, compared to age-matched men, does not represent a sign of comparatively higher CV risk. The uncertainty of measuring NLR in premenopausal women for CV predictive assessment, is even higher because of its oscillating course. Namely, NLR values follow significant variations in hormonal status during particular periods of womens’ life, such as puberty, pregnancy, and menopause, and even variations in the menstrual cycles, which are also periods when ADs are more likely to emerge [28].

### 5.2. Sex Specific Differences in NLR in the Age around Menopause

A sudden drop in NLR values in women at age of menopause and in early post-menopause (age 50–60 years), can be explained by two factors. The first one is a decline in estrogen levels and the loss of estrogens’ effect on neutrophil mobilization from bone marrow to the circulation and their prolonged persistence (Figure 2) [21]. The second one is the shortness of time for the initial accumulation of cardio-metabolic conditions, such as hypertension, MS and DM2, to develop a significant vascular pathology [12,13]. This is partly also caused by a well-preserved endothelial cell function in women at the time of menopause, due to the genetically determined faster resolution of inflammation and the relative resistance to infection in the premenopausal period [43]. At the time of menopause, also, the overall level of inflammation is still low, as it takes time that metabolic and inflammatory changes, that occur with aging, reach the level for increased systemic inflammation [29]. In this regard, it was shown that the level of inflammation increases in parallel to the increase in the degree of insulin resistance, following the path of the transition from the pre-clinical stages of DM2 to DM2 [76].

Microvascular endothelial dysfunction, that develops in the presence of cardio-metabolic conditions such as hypertension and DM2, becomes a significant source of inflammation, by mechanisms such as blood cell aggregation on the surface of endothelial cells, which leads to increased neutrophil activation and recruitment from the circulation to tissue [35]. During this process, also, the activated IL-23/IL-17A immune axis acts in a concert with Ang II in augmenting neutrophil proliferation, differentiation and adhesion to the dysfunctional endothelium [37].

Contrary to the situation in women, a gradual increase in NLR, in men aged 50–60 years, can be explained by only a slow decrease in testosterone levels over time, and the continuation of the effect of testosterone on the same type of immune reaction they had in younger age, which is known as being skewed towards a cell-mediated immunity [21,22]. In a situation of increased CV risk factor accumulation, it is associated with increased inflammation and the progression of atherosclerosis.

When viewed in this way, and as epidemiologic facts indicate, NLR values may be lower in women in the time around menopause, and in early post-menopause, when compared to both, premenopausal women and age-matched men, which makes this parameter unsuitable as a marker of increased CV risk in women. Distinctly from women, a gradual increase in NLR, in middle-aged men (40–60 years old), is likely to reflect the progression of subclinical atherosclerosis, and can therefore be used in CV risk stratification (Figure 2).

### 5.3. Sex Specific Differences in NLR in the Age of Post-Menopause

As indicated by Figure 2, in women in post-menopause, and in men after age 60, NLR steadily increases (Figure 2). These trends can be justified, equally in both sexes, by changes that with aging occur in the immune system, and by the concomitant progression of atherosclerosis and/or end-organ disease, due to the emergence of cardio-metabolic risk factors [21,30,39,40].

The age-related changes in the immune system, which can influence an increase in NLR, include a bias towards myeloid (innate immune) cell differentiation, along with the reduction of lymphopoiesis [30]. Innate immune cells are considered the main source of pro-inflammatory cytokines in individuals of older age [29]. Increased systemic inflammation, and inflammation due to the progression of atherosclerosis and target organ damage, lead to the activation of the IL-23/IL-17A effector and autoimmune pathway, known as being associated with increased neutrophil mobilization and neutrophil-mediated tissue damage [77].

The more slowly rising NLR slope in postmenopausal women compared to age-matched men, as indicated by the longitudinal epidemiologic studies, can be due to a lag in time of the development of significant vascular pathology. In a support to this assumption, epidemiologic facts indicate that men have higher prevalence of CVD than women after the age of 50, and that women catch up with men at the age 70 [7]. In addition, a decline in lymphocyte counts and percentage during the aging process is more marked in men than in women [30].

The genetic predisposition of women for the pro-inflammatory (Th17) immune pathway, and the almost concomitant emergency of hypertension, MS and DM2, that are all conditions associated with microvascular endothelial dysfunction, lead in postmenopausal women to the development of chronic tissue inflammation and a multi-organ disease [24]. These characteristics of immune reaction in postmenopausal women can help explain a female tendency towards non-occlusive IHD and cardiac fibrosis [7,8,9].

Increased availability of self-antigens, due to more intensive tissue damage, and age-related changes of the immune system, which potentiate autoimmune reactions, make that postmenopausal women, especially those with CV comorbidities and the pre-existing end-organ damage and microvascular endothelial dysfunction, are predisposed for late onset ADs [27,30].

We propose that in women after the age 70, who are frail, there may be a more rapid rise in NLR, than in non-frail women (Figure 2). In this terms, the result of the recent experimental study, prepared to show changes in the composition of leukocyte subpopulations in peripheral blood with aging, and between men and women, and how it relates to frailty, has revealed an increase in the blood neutrophil count as the most prominent sign of frailty, and more prominently expressed in women than in men [78].

Frailty is a geriatric condition that attracts much attention of researchers in the recent years because of its influence on prognosis of negative health outcomes [79]. It is manifested with signs of decreased muscle mass and strength, weakness, slow gait speed, a feeling of exhaustion, and low activity, and is considered a result of the reduced homeostatic reserves in multiple physiological systems due to aging and accumulation of age-related diseases. In persons with CVD, the presence of this condition worsens the outcomes [80]. In relation to CV risk prediction, this condition may diminish the prognostic relevance of CV risk factors [81].

An interesting finding, that the rise in NLR values, in women old 70 years and more, does not indicate an increased risk of mortality, is in line with the concept of the “sex-frailty paradox”; this is an observation that although women generally tend to be frail, they live longer than men [82].

Based on the analysis presented above, an increase in NLR values can be used as a sign of increased CV risk yet in women aged 70 years and more, compared to their younger counterparts, and when taking into account the presence of frailty.

## 6. A Summary of the Review

In this review, we defended a hypothesis that NLR, as a marker of increased CV risk and CVD-related mortality, is less efficient in women than in men, which should be taken into account in the population-based studies and predictive models.

This hypothesis is based on the findings of some population-based studies indicating the sex specific differences in the lifetime course of this marker, and on the available knowledge about associations of the hormonal and immune system changes that occur over time, with sex specific differences in the expression of CVD.

With respect to males, their predisposition for atherosclerotic vascular lesions, as the underlying pathology of IHD, and higher prevalence of CVD in middle age and early older age (40–60 years) compared to age-matched women, can be explained by the effect of testosterone on directing the immune reaction towards the predominance of cell-mediated (Th1) responses, and the emergence of CV risk factors such as smoking, hypertension and MS, earlier in life, than it is in females, already in adolescence and early adulthood.

The development of atherosclerosis at age 60 and more, can be considered as a continuation of the progression of subclinical atherosclerosis from middle ages onwards (40–60 years). This is because androgen (testosterone) levels show only a gradual decline with age and are responsible for maintaining the males’ predisposition to Th1 type immune responses, which are in the background of the development of atherosclerotic vascular lesions.

A gradually increasing CV risk, in men after age 60, is associated with CV risk factors emerging at higher rates, and increased levels of inflammation, due to aging and age-related changes of the immune system. The upward-sloping line of NLR reflects well these trends and a CV risk which rises in due course of a lifetime, from middle ages onwards (Figure 2).

In women, the situation with NLR as a marker of increased CV risk is more complex than in men. The lifetime course of the NLR slope is more oscillating and increased NLR values do not reflect well increased CV risk, except in women old 70 years and more, but a distinction needs to be made between frail and non-frail women (Figure 2).

High NLR values in premenopausal women, that are comparatively higher than in men, have their roots in the effects of the female sex hormones, estrogens, on the immune system, and the female genetic predisposition to ADs, both effects prioritizing the role of neutrophils in an inflammatory reaction.

Following a decline in estrogen levels, during peri-menopause and early post-menopause (age range 49–59 years), there is a drop in NLR values. A recovery of the NLR slope begins after the age of 60, simultaneously with the emergence of CV risk factors. In women, CV risk factor patterns typically include hypertension associated with MS and with concomitantly expressed DM2—all disorders that are based on microvascular endothelial dysfunction and chronic tissue inflammation in multiple organs. On such pathological background, the genetically favored pattern of immune reaction, including the IL-23/IL-17A immune pathway, works in a concert with the activated tissue renin-angiotensin system in developing a multi-organ disease.

Age-related changes in the immune system, including a bias towards non-specific and innate immune responses, and the stimulation of innate cell receptors by self-antigens, make older females additionally predisposed to autoimmune reactions and chronic tissue inflammation.

These characteristics of immune reaction in postmenopausal women can help explain the female sex’s tendency toward non-occlusive IHD and cardiac fibrosis.

## 7. Conclusions

This review presents the basis for a better understanding of sex and gender differences in the mechanisms of CVD. The development of CVD relies on the complex interplay between many factors, including genetic, behavioral, metabolic, vascular, hormonal, and immune-related ones. The patterns of these factors change over time and with aging. In particular, the pathophysiological background of CVD in women is poorly understood due to the under-representation of females in experimental and clinical studies. In addition, the difficulty of distinguishing biological and psychological and behavioral influences on female CV health is largely underestimated. Based on the current knowledge and this analysis, NLR, which was shown to have good prognostic properties as a CVD risk marker in men and the general population, seems to have less predictive significance in women. This is due to the higher variability of NLR across their life course. To better evaluate the importance of NLR for CV and mortality risk prediction in women, it is necessary to identify reference and threshold values for specific age groups, to take some periods of life into account, which should include at least pre-menopause, the time around menopause, early post-menopause, and late post-menopause, and to distinguish between older women who are frail and those who are not.

## Figures and Tables

**Figure 1 biomolecules-11-00528-f001:**
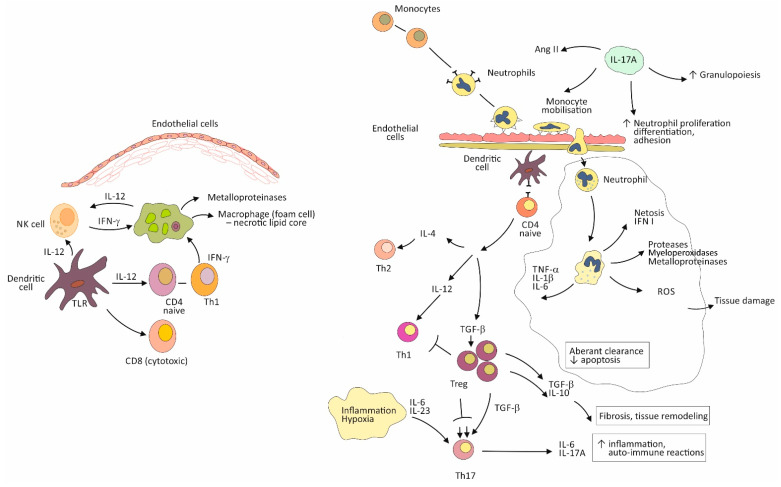
Immune mechanisms included in the development of cardiovascular disease that could explain variations in the neutrophil-to-lymphocyte ratio.

**Figure 2 biomolecules-11-00528-f002:**
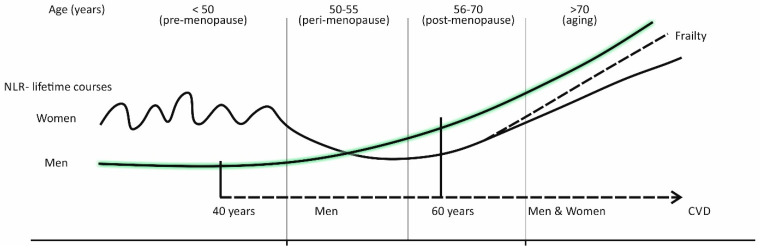
The sex specific lifetime courses of the inflammatory marker “neutrophil-to-lymphocyte ratio” (NLR).

**Table 1 biomolecules-11-00528-t001:** Published papers in which neutrophil-to-lymphocyte ratio (NLR) was assessed in the context of atherosclerotic cardiovascular disease (CVD).

Authors	Findings
Corriere et al. [53]	Demonstrated that NLR is a strong predictor of the presence and number of carotid atherosclerotic plaques
Li et al. [54]	Demonstrated an association between NLR and mixed and non-calcified plaques in the coronary arteries of patients with chest pain
Kaya et al. [55]	Found significantly higher NLR values in patients with severe coronary atherosclerosis
Kalay et al. [56]	Demonstrated that NLR predicts coronary atherosclerosis progression and suggested it as a marker for monitoring
Erturk et al. [57]	NLR values higher than 3.0 were predict CV mortality in patients with peripheral arterial occlusive disease
Tonyali et al. [58]	NLR values equal to or higher than 2.5 were shown to predict severe atherosclerosis with a sensitivity and specificity of 62% and 69%, respectively
Zazula et al. [59]	Found that patients with chest pain that was not caused by cardiac disease, had NLR values of 3.0 ± 1.6, those with chest pain caused by unstable angina had NLR values of 3.6 ± 2.9, and those with MI had much higher values—4.8 ± 3.7 with non-STEMI, and 6.9 ± 5.7 with STEMI, and concluded that NLR value above 5.7 had 91% specificity for the diagnosis of ACS.

**Table 2 biomolecules-11-00528-t002:** Published papers in which NLR was assessed in the context of other cardio-metabolic conditions.

Authors	Findings
Demir et al. [61]	Increased NLR was also found in patients with hypertension of the sort called “non-dipper hypertension”, which does not show a circadian rhythm and is connected with an increased risk of CV events, as a consequence of microvascular changes
Tonyali et al. [58]	Demonstrated in patients with partial or complete nephrectomy, that NLR can represent renal function and renal reserve, making it a good marker of declining renal function
Buyukkaya et al. [62]	Found that increased values for neutrophils and NLR, with an optimal NLR threshold of 1.84, correlate with the severity of MS, without a significant change in the lymphocyte count
Bahadir et al. [63]	Did not find NLR to be a good predictor of inflammation severity in obese patients with MS and without DM2 but indicated that a more significant role was played by CRP and lymphocyte count
Babio et al. [64]	Higher baseline neutrophil counts and an increase in neutrophil counts during follow-up, were both independently associated with a risk of MS in people of 55 years or above, and free of CVD. Although these associations were also seen with total WBC and some other leucocyte subpopulations, neutrophils showed the strongest and most consistent associations, in particular when predicting dyslipidemia associated with MS
Wan et al. [65]	Higher baseline neutrophil counts and an increase in neutrophil counts during follow-up, were both independently associated with a risk of MS in people of 55 years or above, and free of CVD. Although these associations were also seen with total WBC and some other leucocyte subpopulations, neutrophils showed the strongest and most consistent associations, in particular when predicting dyslipidemia associated with MS
Ardahanli et al. [66]	Detected increased NLR values in pre-diabetic patients, compared to healthy controls, suggesting its suitability as a screening marker
Howard et al. [67]	Multiple demographic and lifestyle factors are associated with NLR, independent of important comorbidities, including heart disease, cancer, diabetes, and hypertension

## Data Availability

No applicable.

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
