# Peer review of "Neutrophil-to-Lymphocyte Ratio as a Cardiovascular Risk Marker May Be Less Efficient in Women Than in Men"

_biomolecules, 2021, doi:10.3390/biom11040528_

Round 1

Reviewer 1 Report

The manuscript entitled “Neutrophil-to-lymphocyte ratio as a cardiovascular risk marker in Women” by Dr. Trtica Majnaric et al., explored the neutrophil-to lymphocyte ratio (NLR) in the context of cardiovascular disease in women and the underlying mechanisms. The review is well structured, although there are few issues that should be addressed to improve it.

Major issues

  • Although the authors intend to explore the NLR in the context of CVDs and the possible usefulness as CVD risk marker in women, a huge part of the review is focused on inflammation, immune function, and their changes in relation to the hormonal status. For instance, section 1.2, 1.3 and 1.4 are too long and should be shortened in order to have a major focus on the next sections, the 2 and the 4 (titled “Netrophil-to-lymphocyte radio as a new cardiovascular risk marker). At this regard, the current section 3 (“the role of leukocyte subtypes In inflammation and immune reaction”) should be summarized because it is extremely long and the reader lose the focus. The final section, the number 4 (titled “sex specific differences in NLR across the life course …”), although interesting is too long. While 4.1 is good, 4.2, 4.3 and 4.4 are too long and should be shortened. In fact, there is too information about the influence of sex hormones on immune cell polarization and their responses. Since the review is titled “Neutrophil-to-lymphocyte ratio as a cardiovascular risk marker in Women”, the major topic should be the NLR, variations in relation to sex/moment of life (pre-menopause, menopause, post-menopause and so on), and possible explanations about the differences. In relation to the whole text, these topics are addressed only in few parts of the text.
  • The summary (4.4) or the conclusions should better reflect what is hypothesized by the authors, which is not really clear from the text.

Minor issues

  • The text should be checked because some sentences are too long and difficult to follow.
  • Please read carefully the titles of the sections. For instance, there are two number 3 sections, and two 4.3 sub-sections.

Author Response

Answers to the reviewers` comments

Dear Editor and Reviewers of Biomolecules J,

Thank you for giving us a good guide for making a revision of the manuscript entitled “Neutrophil-to-Lymphocyte Ratio as a Cardiovascular Risk Marker in Women”, subscribed by Lj. Trtica Majnarić and coll.

Actually, to be closer to the changed content of the manuscript, we have changed the title of the paper into “Neutrophil-to-Lymphocyte Ratio as a Cardiovascular Risk Marker May Be Less Efficient in Women Than in Men”.

Overall, we have prepared the paper revision according to your comments.  This included the following changes:

Changes made according to comments of the reviewer 1:

  • We put the focus on the sections engaged in the problems of “Netrophil-to-lymphocyte radio as a new cardiovascular risk marker” and “Sex specific differences in NLR across the life course”, and revised and comprehend sections that describe inflammation, immune function, and their changes in relation to the hormonal status.
  • In this terms, we also changed the order of some sections, and renumbered them.
    • The numeration of the prior section 1. and subsections 1.1. and 1.2. – is left the same
    • The prior subsection 1.3. has been joined to the prior subsection 4.2. – and it is now a new subsection 2.1.
    • The prior subsection 1.4. was threw out.
    • The prior subsections 4.2. – 4.4. (the subsections 4.3. and 4.4. wrongly double marked as 4.3.) are now placed under the section 2, entitled as “Proposed Underlying Mechanisms of Differences Between Men and Women in Cardiovascular Disease Patterns”, and are marked as subsections1. – 2.3.
    • Accordingly, we cleared out the prior Table 1.
    • The prior section 2., entitled “Leukocyte Counts and Ratios as Markers of Inflammation in Cardiovascular Disease” is now numbered as the section 3. The prior section 3., entitled as “The Role of Leukocyte Subtypes in Inflammation and Immune Reaction”, was cleared out, with some parts of the text being incorporated in the current section 2.
    • The prior section 3. (wrongly numbered as No. 3, in fact the section No. 4) – it is now the section 4. – with the same title as it was in old version of the paper, that is: “Neutrophil-to-Lymphocyte Ratio (NLR) as a new Cardiovascular Risk Marker”.
    • According to the comment of the review 2
    • we summarized citations in this section as tables (now Table 1 and 2).
    • The section entitled “Sex Specific Differences in NLR Across the Life Course - Associations with Immune Mechanisms Contributing to Sex Specific Differences in Patterns of CVD”, previously numbered as the section 4, is now section 5. The subsections 4.2. – 4.4., as mentioned above, were revised and replaced, and they are now under the section numbered as the section 2. Instead, three new subsections, numbered as 5.1., 5.2., and 5.3., were included in the section 5, to better evaluate the major topic of the manuscript, namely, the NLR variations in relation to sex moment of life (pre-menopause, menopause, and post-menopause).
    • Accordingly, the Figure 2 (The sex specific lifetime courses of the inflammatory marker “neutrophil-to-lymphocyte ratio” (NLR)), was revised, so that the textual parts were cleared out.
    • A summary of the review (previously numbered as the subsection 4.4.) is now section 6., and it has been revised, so that more clearly states of what we hypothesized in the manuscript.  
    • The section Conclusions, previously numbered as the section 5, is now section 7.

In addition to these changes, we removed some details from Figure 1, to make it more understandable for the readers, and made some changes in the Abstract of the paper, to better reflect the structure and the content of the revised paper. We have also prepared a Graphical Abstract, which will appear on the journal website.

Reviewer 2 Report

It is an interesting and well-written manuscript on cardiovascular diseases in terms of gender and changes in inflammatory activity using a neutrophil-to-lymphocyte ratio. The first part deals with possible pathomechanisms of gender differences in cardiovascular diseases with a focus on vascular dysfunction and atherosclerosis. In the second part, the authors focus on the NLR ratio. The paper summarizes knowledge from several aspects (gender, age, genetic factors, etc) in the form of the Figure and Table, as well as a separate summary at the end of the work.

I have several comments:

The section of this review “3. Neutrophil-to-Lymphocyte Ratio as a new…” (p.8) should be part 4. (part 3. is noted as “The role of leucocyte…” – p. 6). The numerical marking of the individual parts should be revised.

The revision of the citations is necessary for the section related to NLR as a new cardiovascular risk marker (p. 8-10). Some citations are missing, some citations are incorrect, e.g. Corriere and coll., Li and coll., Kaya and coll., Zuzula and coll., etc. The citations should be consistent with references at the end of the work. Moreover, I recommend using “author et al.”, not “author and coll.” Please unify the form of the References.

I recommend summarizing the results from this section regarding NLR and various diseases including CVD in the Table. It would be clearer for the reader.

Author Response

(The authors gave the same response as above.)

Round 2

Reviewer 1 Report

I think that after the revisions, the manuscript is suited for publication.